# Teicoplanin—A New Use for an Old Drug in the COVID-19 Era?

**DOI:** 10.3390/ph14121227

**Published:** 2021-11-26

**Authors:** Vladimir Vimberg

**Affiliations:** Institute of Microbiology, Czech Academy of Sciences, Průmyslova 595, 252 50 Vestec, Czech Republic; vladimir.vimberg@gmail.com

**Keywords:** teicoplanin, SARS-CoV2, dalbavancin, antibiotic, lipoglycopeptide antibiotic, bacteria, COVID-19, co-infection

## Abstract

Teicoplanin is an antibiotic that has been actively used in medical practice since 1986 to treat serious Gram-positive bacterial infections. Due to its efficiency and low cytotoxicity, teicoplanin has also been used for patients with complications, including pediatric and immunocompromised patients. Although teicoplanin is accepted as an antibacterial drug, its action against RNA viruses, including SARS-CoV2, has been proven in vitro. Here, we provide a thorough overview of teicoplanin usage in medicine, based on the current literature. We summarize infection sites treated with teicoplanin, concentrations of the antibiotic in different organs, and side effects. Finally, we summarize all available data about the antiviral activity of teicoplanin. We believe that, due to the extensive experience of teicoplanin usage in clinical settings to treat bacterial infections and its demonstrated activity against SARS-CoV2, teicoplanin could become a drug of choice in the treatment of COVID-19 patients. Teicoplanin stops the replication of the virus and at the same time avoids the development of Gram-positive bacterial co-infections.

## 1. Introduction

The ongoing COVID-19 pandemic requires extraordinary efforts to combat the disease [1]. Several strategies have been proposed: vaccination [2], the usage of virus-neutralizing antibodies [3], and drugs that prevent and/or stop infection [4]. Vaccination strategies are supposed to provide populations with a defense against SARS-CoV2 infection. The usage of neutralization (blocking) antibodies may prevent the virus from entering the human body. Drugs can be used in different stages of an infection to stop and to prevent the further spread of the virus inside the body. The repurposing of available drugs with a known history of medical application might be a useful strategy in combating COVID-19 once they demonstrate effective activity against SARS-CoV2. In this sense, teicoplanin may be one of the best candidates to be repurposed for usage in patients suffering from COVID-19. In this review, we summarize data that can support teicoplanin usage in the treatment of COVID-19 patients.

## 2. Teicoplanin Structure

Teicoplanin was reported for the first time in 1978 and was extracted from *Actinoplanes teichomyceticus*, which was isolated from a soil sample collected in Nimodi Village, Indore, India [5]. The structure of teicoplanin was solved in 1984 [6,7]. It was demonstrated that teicoplanin is a lipoglycopeptide antibiotic (see Figure 1). The antibiotic is formed from a nonribosomal heptapeptide made up of seven aromatic amino acids tailored with sugar residues and a lipid chain. It is produced by bacteria as a mixture of five similar compounds that differ between each other in their fatty acid side chains [8].

## 3. Teicoplanin Usage in Medicine

Teicoplanin was approved in Europe in 1988, two years after its first application to treat bone and soft tissue infections, endocarditis, pneumonia, and sepsis [9]. Today, teicoplanin is sold under the name TARGOCID and is available in many countries around the world (http://www.drugs.com/international/targocid.html (accessed on 12 May 2021)). Teicoplanin has been used in the treatment of serious infections caused by Gram-positive bacteria. It has been used as an alternative antibiotic to vancomycin, which was the first glycopeptide antibiotic approved for usage in 1958. Both antibiotics bind to the D-alanyl-D-alanine (D-Ala-D-Ala) terminus of the bacterial cell wall peptidoglycan precursor. This interaction inhibits peptidoglycan polymerization and subsequent cross-linking steps, resulting in the cessation of cell wall synthesis (see Figure 2). In addition to binding to cell wall precursors, it has been proposed that teicoplanin attaches to cell membrane lipid II substrate through its hydrophobic tail, bringing the antibiotic in the vicinity of the nascent peptidoglycan (however, this has still not been fully confirmed) [10,11].

Teicoplanin demonstrated exceptionally good efficiency in the treatment of infections in different organs (see Table 1, Appendix A). The analysis of the data regarding teicoplanin efficiency showed that treatment with teicoplanin failed in only 16% of cases, mainly due to the misdiagnosis of the infection caused by Gram-negative bacteria. In the rest of the cases, treatment led to either a complete cure (67%) or an improvement (17%). The data were collected from publications available from 1986 to 2020 (a database of the articles describing teicoplanin usage in the treatment of various infections is available in Appendix A). It is of special note that teicoplanin has been successfully applied to treat infections in the respiratory tract, indicating that teicoplanin can reach associated infection sites in the respiratory tract. One of the main advantages of teicoplanin is that it has a relatively long half-life of 30 h [12]. In the majority of cases, it is applied intravenously; however, intramuscular administration is also possible.

## 4. Teicoplanin Distribution in the Human Body

Teicoplanin dosage has undergone a big change from the time of the first application in humans (see details in Appendix A). Today, in the case of serious infections involving deep-seated infections and/or severe infections, a 10–15 mg/kg dosage is recommended to be administered three times every 12 h, with a subsequent maintenance dosage every 24 h [12,13,14]. In the case of mild infections, a 6 mg/kg dosage can be administered [12,15]. The maintenance dosage should be correlated with the parameter of serum albumin; this is because teicoplanin binds to the albumin in blood [16]. A decreased level of serum albumin is correlated with the lower trough concentration of teicoplanin. Another important parameter is renal function because up to 95% of teicoplanin is eliminated by this path [17,18].

Today, it is believed that a stable blood trough concentration can be achieved in the majority of patients on day 2–3 after the initial administration of teicoplanin (reviewed by Pea et al., 2020) [12]. In the case of a 10–15 mg/kg dosage, the trough concentration varies in the range of 23–94 µg/mL [19].

Teicoplanin is not equally distributed all over the body (see Table 2, Appendix A). The most efficient site of teicoplanin accumulation is the heart [20]. In the heart, a concentration of up to 139.8 µg/g can be reached. However, teicoplanin does not penetrate well into cerebrospinal fluid (CFS) or bone [14]. Teicoplanin penetrates at a 4.9 µg/mL concentration into the epithelial lining fluid that covers the alveoli and the small and large airways [21].

Altogether, one can conclude that the concentration of teicoplanin in blood is about 10 times above the threshold of the resistance breakpoint for staphylococci infections. In the case of staphylococci, the resistance breakpoint according to the EUCAST guidelines is above 2 µg/mL. The concentration of teicoplanin achieved in the epithelial lining fluid [21] is also above the staphylococci resistance breakpoint.

It is generally believed that the distribution of teicoplanin is equal between blood and lungs. However, this supposition is based on a study made on murine and piglet animal models (see Appendix A). In the study performed on murine models, the 5 µg/mL concentration of teicoplanin in lung tissue was achieved 2 h after antibiotic administration [22]. It is of concern that, in this study, the dosage administered to murine models was much higher than that administered to humans. In the study most relevant to human dosages, the teicoplanin administered to piglets [23] achieved the same concentration of the antibiotic in the lungs as in the blood, supporting the belief that the teicoplanin concentrations in the blood and lungs of humans are equal.

## 5. Teicoplanin Associated Side Effects

Teicoplanin is considered an antibiotic with a low level of side effects, which is why it is preferred to vancomycin when immunocompromised or pediatric patients have to be treated [12]. The most common side effects are transient and rarely require a cessation of antibiotic administration. The most frequent side effects are nephrotoxicity, rashes, hearing problems, and fever. The side effects, reported in different studies, are summarized in Table 3 (a more detailed overview can be found in Appendix A).

## 6. Anti-SARS-CoV2 Potential of Teicoplanin

The first indication that glycopeptide antibiotics could be used in the treatment of RNA viral infections came in 1993. It was discovered that the glycopeptide antibiotics kistamycin A and B acted against influenza type A virus [24,25].

Teicoplanin antiviral activity was again reported in 2003. It was shown that teicoplanin can inhibit HIV-1 virus in human CEM cell culture at 17 µM (29 µg/mL) half-maximal effective concentration (EC50) [26]. In 2016, teicoplanin activities against Ebola virus (Zaire strain) and SARS-CoV1 were demonstrated [27,28].

Teicoplanin activity against SARS-CoV1 was demonstrated on SARS-CoV1 pseudovirus, expressing the luciferase gene (see Table 4, Appendix A). The concentration of teicoplanin that was needed to inhibit 50% of the expression of the luciferase gene (IC50) in the HEK239T cell line was 3.67 µM (6.6 µg/mL) [28].

The same strategy was used to prove the activity of teicoplanin against SARS-CoV2 pseudovirus in 2019 [29]. The IC50 concentration of teicoplanin in A549 lung epithelial cells culture was 1.66 µM (2.84 µg/mL), as seen in Table 4.

We recently determined the EC50 of teicoplanin against the virus SARS-CoV2 (not pseudovirus) in Vero E6 cells, which was 15.7 µM (26.8 µg/mL) (manuscript in preparation). Although the EC50 is higher than the IC50 for the SARS-CoV2 pseudovirus, the anti-SARS-CoV2 concentration of teicoplanin is readily achieved in the blood and lungs of humans with a 10–15 mg/kg dosage loading regime (12.23–49.95 µM (20.9–85.38 µg/mL)) [19]. This indicates that teicoplanin has the potential to inhibit the progress of SARS-CoV2 in the human body.

## 7. Teicoplanin Mechanisms of Action against SARS-CoV1 and SARS-CoV2

There are two mechanisms that have been proposed for teicoplanin anti-SARS-CoV activity (see Figure 3).

The first mechanism is the inhibition of the cathepsin L protease through the interaction of the teicoplanin lipophilic moiety with the enzyme (see Figure 3A). This interaction inhibits cathepsin L activity and stops the SARS-CoV release from the late endosome [28,29]. The cathepsin L activity is required to disrupt SPIKE protein and ACE2 receptor interaction inside the late endosome, which is a prerequisite for the virus content to be released into the cytoplasm of the cell [30]. The second mechanism is the inhibition of the activity of the SARS-CoV2 main cysteine protease (SARS-CoV2 3CL Pro) at a 1.6 µM concentration (see Figure 3B). Protease is required for the cleavage of the polyproteins of the coronavirus, releasing the functional proteins required for virus replication [31]. Altogether, these studies showed that teicoplanin targets SARS-CoV2 replication at different stages of infection.

The idea of teicoplanin usage in the treatment of the COVID-19 patients was proposed already in the beginning of the pandemic [32,33]. However, it took half year until the first report of teicoplanin usage in COVID-19 patients was published [34]. Teicoplanin was administered to 21 patients after their admission to intensive care unit. Teicoplanin was given at 6 mg/mL dosage for 3 times every 12 h and continued with maintenance dosage of 6 mg/mL every 24 h for 7–12 days. As a result of the treatment, viral clearance was observed in 40% of patients. Thus, teicoplanin was proposed as potentially active for the treatment of patients with COVID-19 [34,35].

## 8. Bacterial Co-Infections in COVID-19 Patients and the Potential of Teicoplanin Usage

In comparison to influenza, a relatively minor percentage of patients hospitalized with COVID-19 display bacterial co-infection. However, it is generally believed that bacterial co-infection is an important factor that decreases the chances of a patient surviving SARS-CoV2 infection [36]. Similarly to influenza, it has been reported that COVID-19 patients co-infected with bacteria stay longer in hospital and, significantly, have up to 48% mortality [36].

The data about the prevalence of bacterial co-infection in COVID-19 patients vary from 3.1 to 42.2% depending on the location where the studies were performed. The most common bacteria detected in COVID-19 patients are shown in Table 5. The highest prevalence of bacterial co-infection was reported in Switzerland and the lowest was in Barcelona, Spain (see Appendix A). Co-infection by Gram-negative bacteria prevailed in the majority of the studies (see Appendix A), with *P. aeruginosa* being the most often detected in the blood or respiratory tract samples of COVID-19 patients.

The ratio of co-infections caused by Gram-positive bacteria is, in general, lower than Gram-negative bacteria; however, it does form a significant part of bacterial co-infections. For instance, *S. aureus* co-infections have been detected in nearly all bacterial co-infection studies. In several studies, the percentage of Gram-positive bacterial co-infections prevailed over other co-infections.

In a study conducted in the United Kingdom, coagulase-negative staphylococci and *S. aureus* co-infections prevailed in 66.6% and 9.25% of the cases, respectively [37].

In a study conducted at Lyon University Hospital, France, the *S. aureus* and *S. pneumoniae* co-infections formed 69.2% and 23.10% of all bacterial co-infections in COVID-19 patients, respectively [38].

In a recent meta-analysis study of the co-infections performed by Westblade et al., 2021, *S. aureus* and *S. pneumoniae* infections prevailed in 31% and 23% of the cases, respectively [36].

Altogether, it seems that the distribution of bacterial species in co-infections in COVID-19 patients varies and probably depends on the local specificity of the infectious bacteria.

One of the most important questions about co-infections is the time when they occur. It was summarized in the work of Westblade et al., 2021 [36], that less than 4% of patients admitted to hospital had a co-infection detected in the bacterial bloodstream or respiratory tract. The exception was France, where almost 20% of the patients admitted to an intensive care unit had an additional bacterial co-infection. Bacterial co-infections were more often detected in patients that had been hospitalized, especially in intensive care units, where up to 29% of the patients picked up a bacterial infection. In patients hospitalized with COVID-19, bacteria in the respiratory tract were detected in 3–19% of cases, and in intensive care unit patients, the figure was 10–21%. There is a significant difference between bacteria species detected in patients prior to and after hospitalization. *S. aureus* and *S. pneumoniae* were the most common co-infection agents in the blood and respiratory tract of patients before admission to hospital. However, inside the hospital environment, CoNS and enterococci were most commonly detected in the blood infections of the patients, and *p. aeruginosa*, *K. pneumoniae* or *S. aureus* in respiratory tract infections. Bacterial co-infections significantly contributed to respiratory failures and an increased risk of patient death [36].

Although data about the prevalence of bacterial species in co-infections are contradictory, it is obvious that Gram-positive bacteria are important causative agents of co-infections in COVID-19 patients.

Teicoplanin has a particularly good efficiency in the treatment of Gram-positive bacterial infections in different organs (see Table 1). Thus, teicoplanin can be used in COVID-19 patients to treat Gram-positive bacterial co-infections. The administration of teicoplanin will stop/prevent SARS-CoV2 infections based on the proven antiviral activity of the antibiotic and at the same time it will stop/prevent Gram-positive bacterial co-infections in COVID-19 patients.

## 9. Threats to Be Considered in Teicoplanin Usage

The general threat to any antibiotic usage is the development of antibiotic resistance by the bacteria, especially in the era of the COVID-19 pandemic, when the amount of antibiotic usage has increased [39]. Increased usage of antibiotics increases the probability of the selection of resistance.

Resistance to teicoplanin in Gram-positive bacteria has been extensively reviewed [40,41]. In brief, the most common glycopeptide antibiotic resistance in enterococci is due to cell wall reprogramming by enzymes encoded in *vanHAX* gene clusters that produce peptidoglycan precursors containing either D-alanine-D-lactate (D-Ala-D-Lac) or D-alanine-D-serine instead of the dipeptide D-alanine-D-alanine (D-Ala-D-Ala), decreasing the affinity of glycopeptide antibiotics to the peptidoglycan (see Figure 4A). In staphylococci, glycopeptide antibiotic resistance is developed mostly by a series of subsequent mutations in genes involved in cell wall metabolism and stress response, linking the resistance phenotype to cell wall thickening, the misregulation of autolysis, and changes in the cell surface anionic charges; this altogether shelters the cell wall synthesis machinery, which is located in the septum of the cells, from the inhibitory action of antibiotics (see Figure 4B) [10,42].

Resistance to teicoplanin in *S. aureus* (>2 µg/mL minimal inhibitory concentration (MIC)) and coagulase-negative staphylococci (CoNS) (>4 µg/mL) were associated with poor clinical prognosis in patients (reviewed by Blaskovich et al., 2018) [41]. The teicoplanin MICs of the resistant *S. aureus* and CoNS can exceed 16 µg/mL; however, in most cases, these high MIC values were achieved in laboratory-driven resistance selection. Hospital-acquired staphylococci strains with higher than 16 µg/mL MIC for teicoplanin are rare [43,44]. Thus, if the right administration regime of teicoplanin is used, then the concentration of the antibiotic achieved in the blood (23–93.9 µg/mL) is enough to stop the growth of even teicoplanin-resistant staphylococci. Therefore, if teicoplanin is going to be used in COVID-19 patients, then it is extremely important to preserve the correct administration protocol for teicoplanin. This will help to achieve a high concentration of teicoplanin and to avoid the development and spread of teicoplanin-resistant bacteria.

## 10. Novel Semisynthetic Lipoglycopeptide Antibiotics in COVID-19 Treatment

Dalbavancin (see Figure 5A) is a semisynthetic lipoglycopeptide antibiotic. It was approved by the US FDA in 2014 and the EMA in Europe in 2015 for the treatment of acute skin and skin structure infections caused by Gram-positive cocci [45,46].

As well as other glycopeptide antibiotics, dalbavancin binds to terminal D-Ala-D-Ala residues of the nascent peptidoglycan of the bacterial cell wall. Although it is believed that the lipophilic substituent interacts with the cell membrane, no data show the lipophilic substituent to be involved in membrane binding. In contrast, it has been shown to play an essential role in an increased half-life [47] due to it binding to blood serum proteins [48].

The half-life of dalbavancin in humans is over 300 h [47]. This allows for the administration of the antibiotic once a week.

Dalbavancin activity against SARS-CoV2 has been recently demonstrated [49]. Dalbavancin is active against SARS-CoV2 in nanomolar concentrations. The EC50 of dalbavancin against SARS-CoV2 in Vero E6 cells was 12.07 nM, which is 1000 times lower than that of teicoplanin.

The mechanism of dalbavancin activity was proposed to be that it binds to the ACE2 receptor (see Figure 5B), thus blocking SARS-CoV2 interaction with the target eukaryotic cell [49].

Although the effect of dalbavancin on the activity of cathepsin L was demonstrated, the concentration at which the dalbavancin affected the protease (400 µM blocked 40% of the cathepsin L activity) was too high. This suggests that cathepsin L is not the primary target of dalbavancin antiviral activity.

In addition to the high antiviral activity of dalbavancin and its long 300 h half-life, dalbavancin can efficiently penetrate into epithelial lining fluid (ELF) [50]. After a single intravenous administration of 1500 mg of dalbavancin, the concentration in the ELF exceeded 1000 times the anti-SARS-CoV2 EC50 value of the antibiotic (1.9 µg/mL) only 4 h after dalbavancin administration (see Table 6). The concentration in blood varied from 279 to 79 µg/mL between 4 and 168 h after the dalbavancin injection. Thus, the antibiotic concentrations not only exceed the concentration needed for the antiviral activity of dalbavancin, but also exceeded the sensitivity levels of *S. aureus* and *S. pneumoniae*.

It therefore follows that dalbavancin has a strong potential to be applied in COVID-19 patients. It would be enough to administer the antibiotic only once per week in order to avoid the spread of SARS-CoV2 and to stop Gram-positive bacterial co-infections [51].

In addition to dalbavancin, activity against SARS-CoV1 and SARS-CoV2 has been demonstrated for two other clinically accepted semisynthetic lipoglycopeptide antibiotics: telavancin (IC50: 3.49 µM (6.1 µg/mL) in SARS-CoV pseudovirus), oritavancin (IC50: 4.96 µM (8.89 µg/mL) in SARS-CoV pseudovirus), and novel semisynthetic lipogycopeptide derivatives of teicoplanin and vancomycin [52,53]. This shows that the novel semisynthetic lipoglycopeptide antibiotics have strong potential to be further developed into antiviral drugs.

## 11. Conclusions

Teicoplanin, as well as dalbavancin, can be used in the treatment of COVID-19 patients. These are well-characterized medicaments, commonly used to treat serious infections caused by multiple antibiotic-resistant Gram-positive bacteria. Teicoplanin and dalbavancin were proven to block COVID-19 virus replication at concentrations that were achieved in blood and in other tissues of the human body. In the case of dalbavancin, the antibiotic achieves a concentration in the human body which is 1000 times higher than that needed to stop the spread of SARS-CoV2. Taking into consideration that COVID-19 patients have the risk of being co-infected in hospital with Gram-positive bacteria, which can severely affect patient prognosis, teicoplanin and dalbavancin will block virus replication and will avoid any bacterial co-infection of the patients. The correct application of the antibiotic instead of the usage of empiric broad-spectrum antibiotics in COVID-19 patients will benefit patient health as well as prevent the spread of antimicrobial resistance.

## Figures and Tables

**Figure 1 pharmaceuticals-14-01227-f001:**
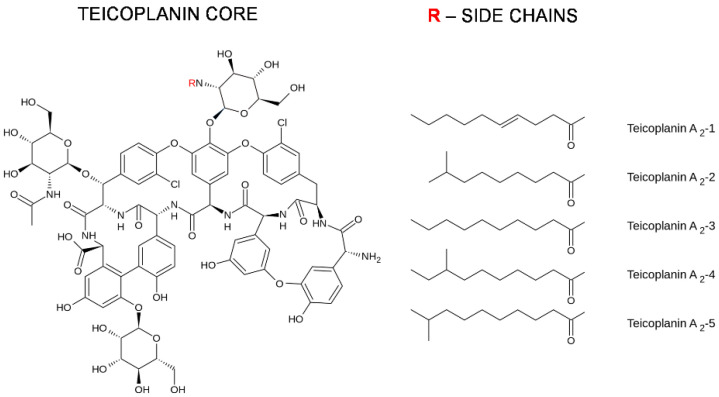
Chemical structure of teicoplanin.

**Figure 2 pharmaceuticals-14-01227-f002:**
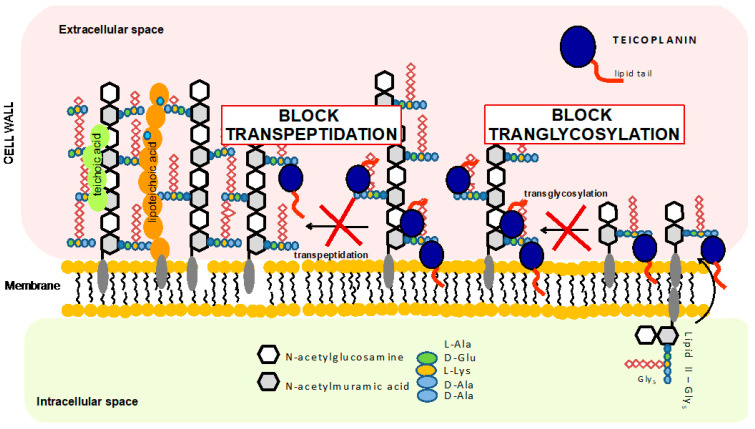
Schematic representation of the teicoplanin-mediated mechanism of inhibition of bacteria cell wall synthesis.

**Figure 3 pharmaceuticals-14-01227-f003:**
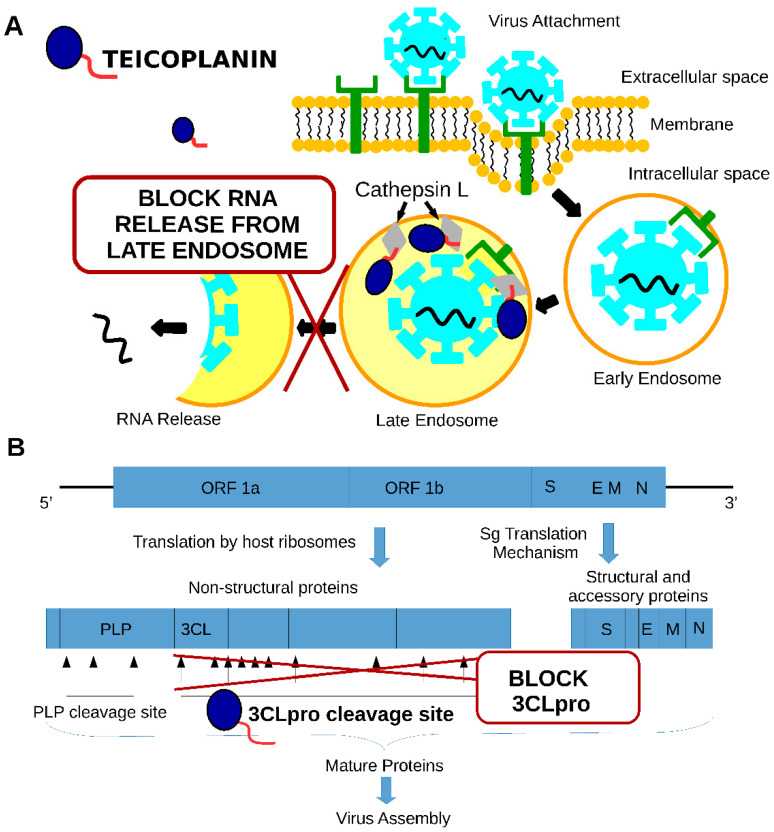
Two mechanisms that have been proposed for teicoplanin anti-SARS-CoV activity: (**A**)—inhibition of cathepsin L activity, (**B**)—inhibition of 3CLpro activity.

**Figure 4 pharmaceuticals-14-01227-f004:**
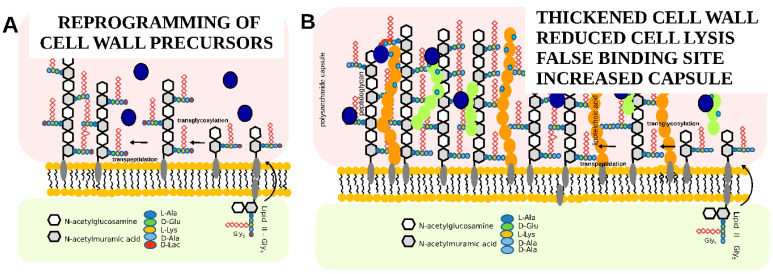
Schematic representation of teicoplanin resistance mechanisms: (**A**)—reprogramming of cell wall precursors, (**B**)—change in the properties of the cell wall.

**Figure 5 pharmaceuticals-14-01227-f005:**
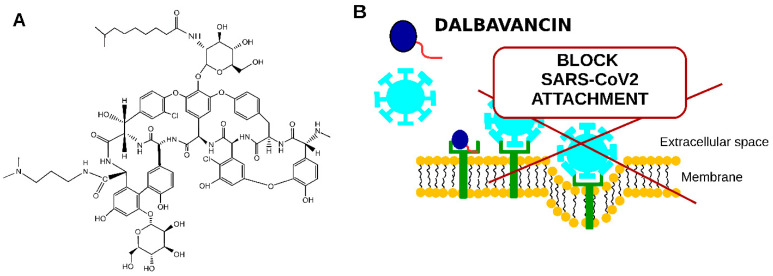
Schematic representation of dalbavancin potential to be used to treat COVID-19 patients. (**A**)—dalbavancin structure, (**B**)—proposed dalbavancin mechanism of anti-SARS-CoV2 activity.

**Table 1 pharmaceuticals-14-01227-t001:** Efficiency of teicoplanin in treatment of infections in different tissues, based on the literature from years 1986 to 2021. Infections in bold highlight the fact that teicoplanin was used to treat respiratory tract.

Infections in	Cured (%)	Improved (%)	Failed (%)
Soft Tissue	76	15	9
Bone/Joint	50	19	31
Septicemia	65	11	24
Lung	68	11	21
Lower Respiratory Tract	73	8	19
Upper Respiratory Tract	73	20	7
Endocarditis	70	5	25
Urinary Tract	76	3	21
Total	67	17	16

**Table 2 pharmaceuticals-14-01227-t002:** Teicoplanin concentrations in different sites of the human body.

Human		**Dosage**	**Concentration**
	Urine	2 or 3 mg/kg (single dose)	22.4 µg/mL
		440 mg (single dose)	25–61 µg/mL
	Bone	400–800 mg (single dose)	1.3–12.7 µg/mL
		10 mg/kg	0.55–25.91 µg/mL
	CFS	400 mg (single dose)	1.3 µg/mL (peak on Day 2)
		10–15 mg/kg	2.1–7.2 µg/mL
	Epithelial Lining Fluid	12 mg/kg	4.9 (2.0–11.8) µg/mL
	Heart	6–12 mg/kg	70.6–139.8 µg/g
	Dialysate	6 mg/kg (single dose)	0.69–1.63 µg/mL
	Skin	400–800 mg (single dose)	1–8.2 µg/mL

**Table 3 pharmaceuticals-14-01227-t003:** Most common teicoplanin side effects.

Side Effects	Frequency (%)
Nephrotoxicity	5.98
Pain in injection site	4.93
Hearing problem	4.76
Drug fever	3.15
Rash	1.68

**Table 4 pharmaceuticals-14-01227-t004:** Teicoplanin effect on SARS-CoV1 and SARS-CoV2.

Year	Virus	Cell Line	IC50 (Luciferase)
2016	HIV-luc/SARS-CoV-S pseudotyped viruses	HEK293T	0.39 µM
2019	2019-nCoV-Spike-pseudoviruses	A549 cells	1.66 μM
Year	Virus	Cell Line	EC50
2021	SARS-CoV-2	Vero E6 cell	15.7 µM

**Table 5 pharmaceuticals-14-01227-t005:** Most common co-infections detected in COVID-19 patients caused by Gram-positive or Gram-negative bacteria.

Gram-Positive Bacteria	Gram-Negative Bacteria
*CoNS*	*P. aeruginosa*
*S. aureus*	*K. pneumoniae*
*S. pneumoniae*	*H. influenzae*
*E. faecium*	*E. coli*
*E. faecalis*	*S. maltophilia*

**Table 6 pharmaceuticals-14-01227-t006:** Dalbavancin concentration after a single 1500 mg administration [50]. SD—standard deviation.

	μg/mL	
Time (Hours)	Plasma (SD)	ELF (SD)
4	279 (32)	1.9 (1.0)
8	222 (27)	3.1 (1.9)
12	194 (24)	3.6 (2.1)
24	169 (20)	2.7 (0.5)
72	120 (14)	7.3 (8.2)
120	94 (11)	11.9 (20.1)
168	79 (9)	2.0 (0.6)

## Data Availability

Data sharing is not applicable to this article.

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
