# Peer review of "Teicoplanin—A New Use for an Old Drug in the COVID-19 Era?"

_pharmaceuticals, 2021, doi:10.3390/ph14121227_

Round 1

Reviewer 1 Report

Teicoplanin is a lipoglycopeptide antibiotic and is used in treating serious gram-positive bacterial infections since 1986. In this manuscript, the authors summarized the therapeutic usage, treatment effects in different tissues, drug distribution in the human body, and the side effects of teicoplanin. Moreover, teicoplanin also exhibits anti-virus activities against Ebola virus, SARS-CoV1, and SARS-CoV2. On the other hand, the authors also discussed the therapeutic potential of teicoplanin in bacterial co-infection of COVID-19 patients. Lastly, the alternative lipoglycopeptide antibiotic such as dalbavancin, telavancin, and oritavancin were discussed in the treatment of COVID-19 patients.

Overall, this article provides a detailed pharmacokinetic, pharmacology, mechanisms for antibiotics and antivirus data of teicoplanin, and provides a thorough guideline for the treatment of SARS-CoV2 or COVID-19.

This manuscript is suitable for publication in Pharmaceuticals.

Major comments

  1. In Table 6 and Section 6, the concentration unit of teicoplanin is molarity. We suggest that the authors provide the concentration both in molarity and weight concentration as it will be more understandable.
  2. Secondary bacterial infections usually occur in patients during or after the initial infection from an infective pathogen such as virus. In addition, secondary bacterial infections could cause morbidity and mortality influenza pandemics and seasonal influenza. We suggest that the authors add some discussion about secondary bacterial infections in this manuscript.

Minor comments

  1. In line 162, the word ‘aboutt’ should be corrected.

Author Response

Response to Review 1 comments:

Point 1 :In Table 6 and Section 6, the concentration unit of teicoplanin is molarity. We suggest that the authors provide the concentration both in molarity and weight concentration as it will be more understandable.

Response 1: The values were added

Point 2: Secondary bacterial infections usually occur in patients during or after the initial infection from an infective pathogen such as virus. In addition, secondary bacterial infections could cause morbidity and mortality influenza pandemics and seasonal influenza. We suggest that the authors add some discussion about secondary bacterial infections in this manuscript.

Response 2: Thank you this note. I agree, that it is very important to consider when co-infection occurs. Short discussion about time of occurrence of the bacterial co-infection in the COVID-19 patient was added. I also tried to mention that in comparison to influenza, the percentage of the bacterial co-infection in the patients prior submission to the hospital is low.

Point 3: Minor comments. In line 162, the word ‘aboutt’ should be corrected.

Response 3: The word was corrected. The whole manuscript was edited by professional language editing service (http://www.proof-edit-copy.com/).

Reviewer 2 Report

The title belies what is included in this manuscript; there is insufficient focus on the potential use of teicoplanin to treat COVID-19 infections. There are no animal or clinical data presented to support the thesis.

It would appear that the paper is focused on what is already known about teicoplanin. If the focus of a revised paper is on the use of teicoplanin in the treatment of bacterial infections, that might be of some benefit, but even then it should be sharply revised to simplify or eliminate most of the tables. For example, in Table 1, data on bacteremia and septicemia could be combined, as could line associated infections, and respiratory infections. Table 2 could be eliminated with ranges of blood concentrations according to dosing stated in the text. Table 3 could also be shortened, by combining skin and fat, and perhaps other tissue sites. Table 4 should be eliminated as there is no need to discuss animal distribution data in the context of this presentation. Table 5 could be of more informational value as a list of side effects according to incidence, as would be in any published pharmacologic data, without any dosages or references.

The section beginning with #6 is of value in the context of the alledged focus of the paper, including Table 6 and Figure 3. However, there is undue focus on bacterial co-infections; those can be presented as a list and frequency of these infections, without need for references. It's uncertain that there is any value in discussion of anti-teicoplanin resistance mechanisms in the context of this presentation.

If the information of dalbavancin is to be included in this presentation, then the title of the paper should be revised to reflect inclusion of other lipoglycopeptide antibiotics, or the title be the potential use of lipoglycopeptide antibiotics in the management of patients with COVID-19 infections.

Author Response

Response to Review 2 comments:

Point 1 : The title belies what is included in this manuscript; there is insufficient focus on the potential use of teicoplanin to treat COVID-19 infections. There are no animal or clinical data presented to support the thesis.

Response 1: I agree that there is not enough data, especially experimental data, to support teicoplanin usage to treat Sars-CoV2 infections. I changed the title from statement to the question. We agree that more study should be done to state that teicoplanin can be used in COVID-19 patients. I also changed the sentence in the abstract that states that teicoplanin activity was proven only in vitro.

Point 2 : It would appear that the paper is focused on what is already known about teicoplanin.

Response 2: It is absolutely right that the review is based on the data available about the teicoplanin. I tried to focus on the possibility of the teicoplanin to be used in COVID-19 patients. There are three aspects that I thought should be considered: previous experience of the teicoplanin usage (generally accepted principle in inhibiting gram-positive bacteria growth, infections treated, concentrations achieved in different organs, side effects), known data about anti-viral activity (in vitro assays that proved activity against viruses, viral mechanisms that were shown to be affected by the teicoplanin, reported experience of the usage of teicoplanin in Italy to treat COVID-19 patients), bacterial co-infections that were reported in COVID-19 patients (gram-positive bacterial infections that can be treated with the teicoplanin and potential risks in teicoplanin usage). In personal communication with the medical doctors in Czech Republic I have realized that in several hospitals, COVID-19 patients are treated with vancomycin to prevent development of the secondary bacterial co-infections. Vancomycin is preferred to teicoplanin because of the lower price. This review, should suggest alternative. I added sentence in section 3 that states that the teicoplanin was introduced as alternative antibiotic to vancomycin. I also added sentence into section 5 that states that vancomycin has stronger side effects than vancomycin. Teicoplanin has similar mechanism of action as vancomycin, but in addition to killing gram-positive bacteria, it can inhibit Sars-CoV2 replication. In the section 6 devoted to bacterial co-infections, I tried to specify that in many cases the co-infection is coming during hospital stay of the patient. Thus, teicoplanin can be used as prevention of the secondary gram-positive bacterial infections.

Point 3 : If the focus of a revised paper is on the use of teicoplanin in the treatment of bacterial infections, that might be of some benefit, but even then it should be sharply revised to simplify or eliminate most of the tables.

For example, in Table 1, data on bacteremia and septicemia could be combined, as could line associated infections, and respiratory infections.

Response 3: I shortened Table 1. The whole overview of data analysis from published articles can be found in supplementary materials. Only the most frequent infections, treated with teicoplanin were left.

Point 4 : Table 2 could be eliminated with ranges of blood concentrations according to dosing stated in the text.

Response 4: Table 2 was eliminated. The table could be found in Supplementary materials.

Point 5 : Table 3 could also be shortened, by combining skin and fat, and perhaps other tissue sites.

Response 5: Table 3 was shortened. The full table could be found in Supplementary materials.

Point 6 : Table 4 should be eliminated as there is no need to discuss animal distribution data in the context of this presentation.

Response 6: I eliminated the table. I think that it is enough to mention this information in the text. Once reader will be interested in the data, it can be found in the Supplementary Materials.

Point 7 : Table 5 could be of more informational value as a list of side effects according to incidence, as would be in any published pharmacologic data, without any dosages or references.

Response 7: Thank you for the suggestions. The table has been changed according your suggestions.

Point 8: The section beginning with #6 is of value in the context of the alledged focus of the paper, including Table 6 and Figure 3. However, there is undue focus on bacterial co-infections; those can be presented as a list and frequency of these infections, without need for references.

Response 8: I decided to leave the part of the bacterial co-infections. I modified the part about co-infections, stressing the fact that bacteria infect patient in hospital. Thus, usage of teicoplanin can be used as prevention of the bacterial co-infection. Table (now Table 5) is left to stress the discrepancy of the data on bacterial co-infection.

Point 9: It's uncertain that there is any value in discussion of anti-teicoplanin resistance mechanisms in the context of this presentation.

Response 9: I think that if the teicoplanin is supposed to be tried in COVID-19 patients, then it is important to be aware of the threats that can come. Antibiotic resistance is one of the most important global threats that should be considered every time, when antibiotic is used. If teicoplanin is going to be used in COVID-19 patients, then it is crucial to keep right dosage regime in order to avoid resistance selection.

Point 10: If the information of dalbavancin is to be included in this presentation, then the title of the paper should be revised to reflect inclusion of other lipoglycopeptide antibiotics, or the title be the potential use of lipoglycopeptide antibiotics in the management of patients with COVID-19 infections.

Response 10: Dalbavancin part has been moved to the section about novel lipoglycopeptide antibiotics. I think that in this way, I keep the main focus of the review on the teicoplanin, but I show that alternatives to the teicoplanin do exist.

Round 2

Reviewer 2 Report

The authors have responded to most of the suggestions of this reviewer. It would still be better to revise Table 5 regarding co-infections as most of the co-infections cold be simply listed without percentages and references.

Author Response

Point 1 :  The authors have responded to most of the suggestions of this reviewer. It would still be better to revise Table 5 regarding co-infections as most of the co-infections cold be simply listed without percentages and references. 

Response 1: Dear Reviewer, You are right, that the Table 5 can be shortened without compromising the text.  I have revised the table, listing the most common gram-positive and gram-negative bacteria infecting COVID-19 patients. The rest of the information can be found in Supplementary Materials.

Point 2 :  English language and style are fine/minor spell check required 

Response 2: Thank you very much for careful review of the manuscript. The manuscript has been edited by professional language editing service (http://www.proof-edit-copy.com/).